# Weather2vec: Representation Learning for Causal Inference with Non-Local Confounding in Air Pollution and Climate Studies

## Abstract

Non-local confounding (NLC) can bias the estimates of causal effects when treatments and outcomes of a given unit are dictated in part by the covariates of other units. This paper first formalizes the problem of NLC using the potential outcomes framework, providing a comparison with the related phenomenon of causal interference. Then it investigates the use of neural networks – specifically U-nets – to address it. The method, termed *weather2vec*, uses balancing scores to encode NLC information into a scalar or vector defined for each observational unit, which is subsequently used to adjust for NLC. We implement and evaluate the approach in two studies of causal effects of air pollution exposure.

## 1 INTRODUCTION

Causal effects of spatially-varying exposures on spatially-varying outcomes may be subject to *non-local confounding* (NLC), which occurs when the treatments and outcomes for a given unit are partly modified by *covariates* of other nearby units [Cohen-Cole and Fletcher, 2008, Florax and Folmer, 1992, Fletcher and Jung, 2019, Chaix et al., 2010, Elhorst, 2010]. In simple cases, NLC can be reduced using simple summaries of non-local data, such as the averages of the covariates over pre-specified neighborhoods [Chaix et al., 2010]. But in many realistic settings, NLC stems from a complex interaction of spatial factors that cannot be easily accounted for using simple *ad hoc* summaries. For such scenarios, this article proposes a representation learning method, termed *weather2vec*, to assist in estimating causal effects in the presence of NLC. *Weather2vec* encodes NLC using a neural network (NN) — specifically a U-net — such that the learned representation can be used for causal estimation in conjunction with standard causal inference

tools. The technique leverages the analogy between rasterized spatial data and images, and it is applicable to broad settings where covariates are available on a grid of units, and the outcome and treatment are (possibly sparsely) observed throughout the same spatial domain. We elaborate connections to the distinct but related phenomenon of *interference*, which could arise when the outcome at a given location depends on *treatments* applied at other locations.

This article has three aims:

1. Provide a rigorous characterization of NLC using the potential outcomes framework [Rubin, 2005]. The connection with interference and some related methods [Tchetgen and VanderWeele, 2012, Forastiere et al., 2021, Sobel, 2006] is also discussed.
2. Expand the library of NN methods in causal inference by proposing a U-net [Ronneberger et al., 2015] as a viable model to account for NLC in conjunction with standard causal inference methods. We investigate two mechanisms to obtain the latent representations: one supervised, and one self-supervised.
3. Establish a promising research direction for addressing confounding in scientific studies of air pollution, climate change, and meteorology, in which NLC driven by meteorology is a common problem for which widely applicable tools are lacking. Two applications will be discussed in detail: the first one estimates the air quality impact of power plant emissions controls; the second one is an application to the problem of meteorological detrending [Wells et al., 2021] to deconvolve climate variability from policy changes when characterizing long-term air quality trends. A simulation study in Appendix A accompanies these examples and compares several alternative approaches.

**Related work**   Previous research has investigated NNs for the (non-spatial) estimation of balancing scores [Keller et al., 2015, Westreich et al., 2010, Setoguchi et al., 2008] and meteorological detrending [Lu and Chang, 2005]. NNs have also been considered to extend beyond estimation of pop-

ulation average treatment effects and target individualized treatment effects (ITE) [Shalit et al., 2017, Johansson et al., 2016, Shi et al., 2019] that seek to estimate counterfactuals for each observed unit based on a structural model. None of these works specifically consider NLC. Direct applications of U-nets in air pollution and climate science include forecasting [Larraondo et al., 2019, Sadeghi et al., 2020] and estimating spatial data distributions from satellite images [Hanna et al., 2021, Fan et al., 2021], indicating that U-nets are powerful tools to manipulate rasterized weather data.

Forms of NLC have been investigated in spatial econometrics. For instance, WX-regression models [Elhorst, 2010] formulate the outcome as a linear function of the treatment and the covariates of some pre-specified neighborhood. Similarly, CRAE [Blier-Wong et al., 2020], which resembles the self-supervised formulation of *weather2vec*, uses an autoencoder to encode pre-extracted patches of regional census data into a lower-dimensional vector that is fed into an econometric regression. In contrast to these approaches, *weather2vec* estimates balancing scores, which have known benefits that include the ability to empirically assess the threat of residual confounding and the offer of protection against model misspecification that arises when modeling outcomes directly [Rubin, 2008].

There is also a maturing literature on adjusting for unobserved spatially-varying confounding [Reich et al., 2021, Veitch et al., 2019, Papadogeorgou et al., 2019]. For methods that rely on spatially-structured random effects, results in [Khan and Calder, 2020] highlight the sensitivity to misspecification of the random effects for the purposes of confounding adjustment. The distance adjusted propensity score matching (DAPSm) method in [Papadogeorgou et al., 2019] foregoes formulation of spatial random effects by matching units based jointly on estimated propensity scores and spatial proximity under the rationale that spatial proximity can serve as a proxy for similarity in spatially-varying covariates. In a context where network proximity is viewed analogously to spatial proximity, Veitch et al. [Veitch et al., 2019] extend this idea further to show that, under certain regularity conditions, network proximity can be used as a proxy for a network-level unobserved confounder. They propose a mechanism to learn embeddings that capture confounding information and used them together with inverse probability weighting [Cole and Hernán, 2008] to obtain unbiased causal estimates. Importantly, Veitch et al. [Veitch et al., 2019] only consider the "pure homophily" case, where the entirety of the confounding is assumed to be encoded by relative position in the network. While some of these methods could be useful for NLC, they all primarily target settings where confounding is local.

Finally, NLC is distinct from, but notionally similar to, *causal interference* [Tchetgen and VanderWeele, 2012, Forastiere et al., 2021, Sobel, 2006, Zigler and Papadogeorgou, 2021]. Both arise from spatial (or network) interaction, and both impose limitations on standard causal inference methods. The distinction between NLC and interference is often acknowledged in the interference literature, although specific methods for NLC and a formal treatment of NLC have been ignored.

## 2 POTENTIAL OUTCOMES AND NLC

The potential outcomes framework, also known as the Rubin Causal Model (RCM) [Rubin, 2008], distinguishes between the observed outcomes $Y_s$ and those that would be observed under counterfactual (potential) treatments. The treatment $A_s$ is assumed to be binary for ease of presentation, although the ideas generalize to other treatments. Some additional notation: $\mathbb{S}$ is the set where the outcome and treatment are measured; $\mathbb{G} \supset \mathbb{S}$ is a grid that contains the rasterized covariates $\boldsymbol{X}_s \in \mathbb{R}^d$; indexing by a set $B \subset \mathbb{G}$ means $\mathbf{X}_B = \{\boldsymbol{X}_s \mid s \in B\}$; $X \perp\!\!\!\perp Y \mid Z$ means that $X$ and $Y$ are conditionally independent given $Z$. Throughout $\boldsymbol{X}_s$ is assumed to consist of pre-treatment covariates only, meaning they are not affected by the treatment or outcome.

**Definition 1** (Potential outcomes). *The potential outcome $Y_s(\boldsymbol{a})$ is the outcome value that would be observed at location $s$ under the global treatment assignment $\boldsymbol{a} = (a_1, \ldots, a_{|\mathbb{S}|})$.*

For $Y_s(\boldsymbol{a})$ to depend only on $a_s$, the RCM needs an additional condition called the *stable unit treatment value assumption*, widely known as SUTVA, and encompassing notions of *consistency* and ruling out *interference*.

**Assumption 1** (SUTVA). *(1) Consistency: there is only one version of the treatment. (2) No interference: the potential outcomes for one location do not depend on treatments of other locations. Together, these conditions imply that $Y_s(\boldsymbol{a}) = Y_s(a_s)$ for any assignment vector $\boldsymbol{a} \in \{0,1\}^{|\mathbb{S}|}$, and that the observed outcome is the potential outcome for the observed treatment, i.e., $Y_s = Y_s(A_s)$.*

The potential outcomes and SUTVA allow to define an important estimand of interest: the average treatment effect.

**Definition 2** (Average treatment effect). *The average treatment effect (ATE) is the quantity $\tau_{ATE} = |\mathbb{S}|^{-1} \sum_{s \in \mathbb{S}} \{Y_s(1) - Y_s(0)\}$.*

One cannot estimate the ATE directly since one never simultaneously observes $Y_s(0)$ and $Y_s(1)$. The next assumption in the RCM formalizes conditions for estimating the ATE, (or other causal estimands) with observed data by stating that any observed association between $A_s$ and $Y_s$ is not due to an unobserved factor.

**Assumption 2** (Treatment Ignorability). *The treatment $A_s$ is ignorable with respect to some vector of controls $\boldsymbol{L}_s$ if and only if $Y_s(1), Y_s(0) \perp\!\!\!\perp A_s \mid \boldsymbol{L}_s$.*

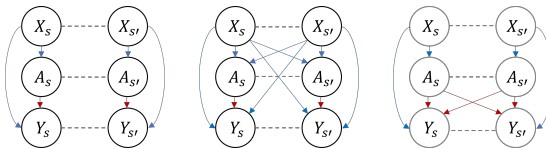

(a) Local Confound-  (b) NLC, no inter-  (c) Interference, no
ing.             ference.         NLC.

Figure 1: Confounding types.

For the sake of brevity, we will say that $\boldsymbol{L}_s$ is *sufficient* to mean that the treatment is ignorable conditional on $\boldsymbol{L}_s$. NLC occurs when local covariates are not sufficient. It is formally stated as follows:

**Definition 3** (Non-local confounding). *We say there is non-local confounding (NLC) when there exist neighborhoods $\{\mathcal{N}_s \subset \mathbb{G} \mid s \in \mathbb{S}\}$ such that $\boldsymbol{L}_s = \mathbf{X}_{\mathcal{N}_s}$ is sufficient and the neighborhoods are necessarily non-trivial ($\mathcal{N}_s \neq \{s\}$).*

Figures 1a and 1b show a graphical representation of local confounding and NLC respectively. Figure 1c shows the distinct phenomenon of interference for contrast. (Additional discussion of interference is in section 4.) Horizontal dotted lines emphasize that there may be spatial correlations in the covariate, treatment and outcome processes that do not result in confounding.

Subsequent discussion of the size of the NLC neighborhood, $\mathcal{N}_s$, will make use of the following proposition stating that a neighborhood containing sufficient confounders can be enlarged without sacrificing the sufficiency.

**Proposition 1.** *Let $\boldsymbol{L}_s$ be a sufficient set of controls including only pre-treatment covariates. and let $\boldsymbol{L}'_s$ be another set of controls satisfying $\boldsymbol{L}'_s \supset \boldsymbol{L}_s$. Then, $\boldsymbol{L}'_s$ is also sufficient.*

All the proofs are in Appendix C. We conclude this section with a classic result stating that any sufficient $\boldsymbol{L}_s$ can be used to estimate the ATE from quantities and relations in the observed data.

**Proposition 2.** *Assume SUTVA holds and that $\boldsymbol{L}_s$ is sufficient. Then*

$$\tau_{ATE} = \mathbb{E}\left[\mathbb{E}[Y_s \mid \boldsymbol{L}_s, A_s = 1] - \mathbb{E}[Y_s \mid \boldsymbol{L}_s, A_s = 0]\right],$$
(1)

*where s is taken uniformly at random from $\mathbb{S}$.*

## 3 ADJUSTMENT FOR NON-LOCAL CONFOUNDING WITH *WEATHER2VEC*

Most commonly, there will be only be one observation for each $s$, and $\mathbb{S}$ can also be small, requiring structural assumptions that enable the identification of causal effects.

A natural structure to consider is spatial stationarity. Intuitively, it entails that the distributions of $Y_s$ and $A_s$ with respect to a neighboring covariate $\boldsymbol{X}_{s'}$ should only depend on $s - s'$ (their relative position). U-nets, originally designed for applications in biomedical image segmentation, provide a practical spatially stationary computational model. They possess the ability to efficiently transform the input grid of covariates $\mathbf{X}_{\mathbb{G}}$ onto an output grid $\mathbf{Z}_{\theta,\mathbb{G}} = f_\theta(\mathbf{X}_{\mathbb{G}})$ of same spatial dimensions such that each $\boldsymbol{Z}_{\theta,s} \in \mathbf{Z}_{\theta,\mathbb{G}}$ localizes contextual spatial information from the input grid.

The essence of *weather2vec* is to define appropriate learning tasks to obtain the weights $\theta$, specified in the form of a loss function or a probabilistic observation model. Two such tasks are considered, summarized below and described in detail in subsequent sections.

1. (**Supervised**) Assuming the treatment and outcome are densely available over $\mathbb{G}$, regress $A_s$ on $\boldsymbol{Z}_{\theta,s}$ (propensity score regression) or $Y_s$ on $\boldsymbol{Z}_{\theta,s}$ (prognostic score regression).
2. (**Self-supervised**) If the treatment and outcome are not densely available over $\mathbb{G}$, then learn $\theta$ so that $\boldsymbol{Z}_{\theta,s}$ is highly predictive of $\boldsymbol{X}_{s'}$ for any $s'$ within a specified radius of $s$.

Appendix B briefly summarieze the U-net computational model. Refer to [Ronneberger et al., 2015] for full details.

### 3.1 LEARNING NLC REPRESENTATIONS VIA SUPERVISED REGRESSION

The supervised approach links the proposed representation learning to the procedure of learning a balancing score [Rubin, 2005, Hansen, 2008] in causal inference. We recall the definition here for completeness.

**Definition 4** (Balancing score). $b(\boldsymbol{L}_s)$ *is a* balancing score *iff $A_s \perp\!\!\!\perp \boldsymbol{L}_s \mid b(\boldsymbol{L}_s)$. The coarsest balancing score is $b(\boldsymbol{L}_s) := \Pr(A_s = 1 \mid \boldsymbol{L}_s)$, widely known as the* propensity score.

**Definition 5** (Prognostic score). $b(\boldsymbol{L}_s)$ *is a* prognostic score *iff $Y_s(0) \perp\!\!\!\perp \boldsymbol{L}_s \mid b(\boldsymbol{L}_s)$. The coarsest prognostic score is $b(\boldsymbol{L}_s) := \mathbb{E}[Y_s(0) \mid \boldsymbol{L}_s]$.*

The propensity score blocks confounding through the treatment [Rubin, 2005]; prognostic scores do so through the outcome [Hansen, 2008]. (Confounders need to be associated with both the treatment and the outcome.) The importance of these definitions is summarized by the next well-known result.

**Proposition 3.** *If $b(\boldsymbol{L}_s)$ is a balancing score, then $\boldsymbol{L}_s$ suffices to control for confounding iff $b(\boldsymbol{L}_s)$ does. The same result holds for the prognostic score under the additional assumption of no effect modification.*

Learning $\theta$ through supervision results in an efficient scalar $\boldsymbol{Z}_{\theta,s}$ compressing NLC information, allowing for $\theta$ to just attend to relevant neighboring covariate information that pertains to confounding. However, supervision may not be possible to use it with small-data studies where $Y_s$ and $A_s$ are only measured sparsely. In such cases, the supervised model will likely overfit to the data. For example, in one motivating application, $\mathbb{S}$ consists only of measurements at 473 power plants, while the size of $\mathbb{G}$ is $128 \times 256$. An over-fitted propensity score would result in insufficient "overlap" [Stuart, 2010] by assigning zero probability to the unobserved treatment, resulting in causal inferences that would rely on model extrapolation to areas where covariate information is not represented in both treatment groups. To avoid this, the self-supervised approach targets scenarios with sparse $\mathbb{S}$.

## 3.2 REPRESENTATIONS VIA SELF-SUPERVISED DIMENSIONALITY REDUCTION

Self-supervision frames the representation learning problem as dimension reduction without reference to the treatment or outcome. The learned representation is then used to learn a balancing score for causal effect estimation in a second analysis stage. This approach requires specification of a fixed neighborhood size $R$ containing the information to be reduced, resulting on different representations for different choices of $\mathcal{N}_s$. In practice, as one of our case studies will demonstrate, one can choose $R$ with standard model selection techniques (such as AIC and BIC) in the second stage of learning a balancing score. The dimensionality reduction's objective is that $\boldsymbol{Z}_{\theta,s}$ encodes predictive information of any $\boldsymbol{X}_{s+\delta}$ for $(s+\delta) \in \mathcal{N}_s$. The dimension $k$ of $\boldsymbol{Z}_{\theta,s}$ is specified as a hyper-parameter depending on the size of $\mathcal{N}_s$ and the dimensionality of $\boldsymbol{X}_s$.

A simple predictive model $\boldsymbol{X}_{s+\delta} \approx g_\phi(\boldsymbol{Z}_{\theta,s}, \delta)$ is proposed. First, let $\boldsymbol{\Gamma}_\phi(\cdot)$ be a function taking an offset $\delta$ as an input and yielding a $k \times k$ matrix, and let $h_\psi(\cdot) \colon \mathbb{R}^k \to \mathbb{R}^d$ be a decoder with output values in the covariate space. The idea is to consider $\boldsymbol{\Gamma}_\phi(\delta)$ as a selection operator acting on $\boldsymbol{Z}_{\theta,s}$. See Appendix D for additional intuition. The model, defined all over the grid $\mathbb{G}$, can be written succinctly as

$$\boldsymbol{X}_{s+\delta} \mid \theta, \phi, \psi, \Sigma \sim \mathrm{Normal}(h_\psi(\boldsymbol{\Gamma}_\phi(\delta)\boldsymbol{Z}_{\theta,s}), \Sigma) \quad (2)$$

$\forall s \in \mathbb{G}, \ \forall \delta \colon \|\delta\| \leq R$. The negative log-likelihood loss function can be optimized using stochastic gradient descent by sampling $\delta$ uniformly from the unit ball of radius $R$. A connection with PCA appears in Appendix E.

## 4 NLC AND INTERFERENCE

To ground the discussion, we will focus on the version of neighborhood-level interference described by Forastiere et al. [Forastiere et al., 2021], formalized with the *stable unit neighborhood treatment value assignment* (SUTNVA):

**Assumption 3** (SUTNVA). *(1) Consistency: there is only one version the treatment. (2) Neighborhood-level interference: for each location $s$, there is a neighborhood $\mathcal{N}_s$ such that the potential outcomes depend only on the treatments at $\mathcal{N}_s$. Together, these conditions imply that $Y_s(\boldsymbol{a}) = Y_s(\boldsymbol{a}_{\mathcal{N}_s})$ for any assignment vector $\boldsymbol{a} \in \{0,1\}^{|\mathbb{S}|}$, and that the observed outcome is the potential outcome for the observed treatment, i.e., $Y_s = Y_s(\mathbf{A}_{\mathcal{N}_s})$.*

In many cases it may be expected that interference and NLC occur together, and Figure 1 indicates how NLC can be confused with SUTNVA when not accounting for the neighboring covariates by inducing spurious correlations between a unit's outcome and other units' treatments. While the presence of interference generally requires specific techniques to account for spill-over effects when estimating quantities such as the ATE, [Forastiere et al., 2021] offer specialized conditions that can lead to unbiased estimation of one type of causal effect defined under SUTNVA. We summarize these conditions in the following proposition, also encompassing the definition of the direct causal effect, which is one estimand of interest (alternative to the ATE) that arises in the presence of interference. The proposition requires potential outcomes of the form $Y_s(a_s = a, \mathbf{A}_{\mathcal{N}_s \setminus \{s\}})$, which are a short-hand notation for $Y_s(a_s = A_s, \{a_{s'} = A_{s'}\}_{s \in \mathcal{N}_s \setminus \{s\}})$, which assigns all the neighbors of $s$ to their observed treatments in the data.

**Proposition 4.** *Assume SUTNVA and define the direct average treatment effect (DATE) as*

$$\tau_{DATE} = |\mathbb{S}|^{-1}\sum_{s \in \mathbb{S}}\{Y_s(a_s = 1, \mathbf{A}_{\mathcal{N}_s \setminus \{s\}})$$
$$- Y_s(a_s = 0, \mathbf{A}_{\mathcal{N}_s \setminus \{s\}})\}.$$

*Then if (1) $\mathbf{A}_{\mathcal{N}_s} \perp\!\!\!\perp (Y_s(\boldsymbol{a}))_{\boldsymbol{a} \in \{0,1\}^{|\mathcal{N}_s|}} \mid \boldsymbol{L}_s$ and (2) $A_s \perp\!\!\!\perp A_{s'} \mid \boldsymbol{L}_s$ for all $s \in \mathbb{S}, s' \in \mathcal{N}_s$. Then*

$$\tau_{DATE} = \mathbb{E}\left[\mathbb{E}[Y_s \mid \boldsymbol{L}_s, A_s = 1] - \mathbb{E}[Y_s \mid \boldsymbol{L}_s, A_s = 0]\right].$$

Conditions (1) and (2) correspond, respectively, to the notions of neighborhood-level ignorability and conditional independence of the neighboring treatments. The proposition states that the same estimator of $\tau_{\mathrm{ATE}}$ in the non-interference case yields unbiased estimates of $\tau_{\mathrm{DATE}}$. The simplest scenario satisfying these conditions is represented in Figure 1c; adjusting only for local covariates ($\boldsymbol{L}_s = \{X_s\}$) is sufficient.

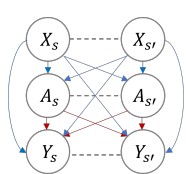

Figure 2: Interference + NLC.

The presence of NLC can violate conditions (1) and (2) of Proposition 4. To see this, consider Figure 2 representing the co-occurrence of interference and NLC. Adjusting only for local covariates would violate neighborhood ignorability condition (1) with a spurious correlation between $Y_s$ and

$A_{s'}$ (through the backdoor path $Y_s \leftarrow \mathbf{X}_{s'} \rightarrow A_{s'}$). Similarly, a spurious correlation between $A_s$ and $A_{s'}$ would persist (via the the path $A_s \leftarrow \mathbf{X}_{s'} \rightarrow A_{s'}$). For such cases, *weather2vec* can play an important role in satisfying (1) and (2) since, after controlling for NLC (consisting in Figure 2 of adjusting for both $\mathbf{X}_s$ and $\mathbf{X}_{s'}$ and blocking the incoming arrows from neighboring covariates into one's treatments and outcomes), the residual dependencies would more closely resemble those of Figure 1c. In summary, adjusting for NLC with *weather2vec* can aid satisfaction of the conditional independencies required to estimate causal effects with the same estimator used to estimate the ATE absent interference.

# 5 APPLICATIONS IN AIR POLLUTION

**Application 1: Quantifying the impact of power plant emission reduction technologies** Because many air quality regulations are inherently regional and certain types of plants are concentrated in regions with similar weather and economic demand factors (e.g., those in most need of emissions controls), regional weather correlates with the assignment of the intervention. Further, weather patterns dictate regional differences in the formation and dispersion of ambient air pollution. Thus, the weather is a potential confounding factor, but one for which the relevant features are inherently regional and may not be entirely characterized at a given location by point measurements of covariates including local wind, temperature, precipitation, etc..

*Self-supervised features from NARR.* We downloaded monthly NARR data [Mesinger et al., 2006] containing averages of gridded atmospheric covariates across mainland U.S. for the period 2000-2014. We considered 5 covariates: temperature at 2m, relative humidity, total precipitation, and north-sound and east-west wind vector components. For each variable we also include its year-to-year average. Each grid cell covers roughly a $32 \times 32$ km area and the lattice size is $128 \times 256$. We implemented the self-supervised *weather2vec* with a lightweight U-net of depth 2, 32 hidden units, and only one convolution per level. See Appendix H for more details and schematic of the U-net architecture. To measure the quality of the encoding, Figure 3a shows the percentage of variance explained ($R^2$), comparing with neighbor averaging and local values. The results shows that the 32-dimensional self-supervised features provide a better reconstruction than averaging and using the local values. For instance, the 300km averages only capture 82% of the variance, while the self-supervised *weather2vec* features capture 95%. See Appendix I for details on the calculation of the $R^2$ and neural network architecture.

*Estimated pollution reduction.* We evaluate different propensity score models for different neighborhood sizes of the June 2004 NARR *weather2vec*-learned features with the same logistic model and other covariates as in DAPSm, aug-

mented with the self-supervised features. We selected the representation using features within a 300km radius on the basis of its accuracy, recall, and AIC in the propensity score model relative to other considered neighborhood sizes (Figure 3b). The causal effects are then obtained by performing 1:1 nearest neighbor matching on the estimated propensity score as in DAPSm. Figure 3c compares treatment effect estimates for different estimation procedures. Overall, standard (naive) matching using the self-supervised features is comparable to DAPSm, but without requiring the additional spatial adjustments introduced by DAPSm. The same conclusion does not hold when using local weather only , which (as in the most naive adjustment) provides the scientifically un-credible result that emissions reduction systems significantly *increase* ozone pollution. Do notice the wide confidence intervals which are constructed using conditional linear models fit to the matched data sets [Ho et al., 2007]. Thus, while the mean estimate shows a clear improvement, the intervals shows substantial overlap, warranting caution.

**Application 2: Meteorological detrending of sulfate** We investigate meteorological detrending of the U.S. sulfate ($SO_4$) time series with the goal (common to the regulatory policy and atmospheric science literature) of adjusting long-term pollution trends by factoring out meteorologically-induced changes and isolating impacts of emission reduction policies [Wells et al., 2021]. We focus on $SO_4$ because it is known that its predominant source in the U.S. is $SO_2$ emissions from coal-fired power plants, on which observed data are available for comparison. Thus, we hypothesize that an effectively detrended $SO_4$ time series will closely resemble that of the power plant emissions.

*Prognostic score.* We obtained gridded $SO_4$ concentration data publicly available from the Atmospheric Composition Analysis Group [Group, 2001, van Donkelaar et al., 2021], consisting of average monthly value for each raster cell in the mainland U.S. for the period of study 2000–2014. The data is aggregated into 32km-by-32km grids to match the resolution of atmospheric covariates. The model uses a U-net with a Gaussian likelihood (quadratic loss). Since the prognostic score is defined based on outcome data in the absence of treatment, we leverage the fact that the power plant emissions were relatively constant for the period 2000-2006 and consider this period as representing the absence of treatment. The model predictions, aggregated by all points in the grid is shown in Figure 4a. The difference between the red line (the prognostic score fit) and the black dotted line (the $SO_4$) observations during 2000 - 2006 is a proxy for the meteorology-induced changes in the absence of treatment.

*Trend estimation.* For comparability we adhere to the recommended detrending model by [Wells et al., 2021]. Accordingly, we specify a regression with a year and seasonal fixed-effect term. Rather than pursue an entirely new methodology for detrending, we intentionally adhere to standard best prac-

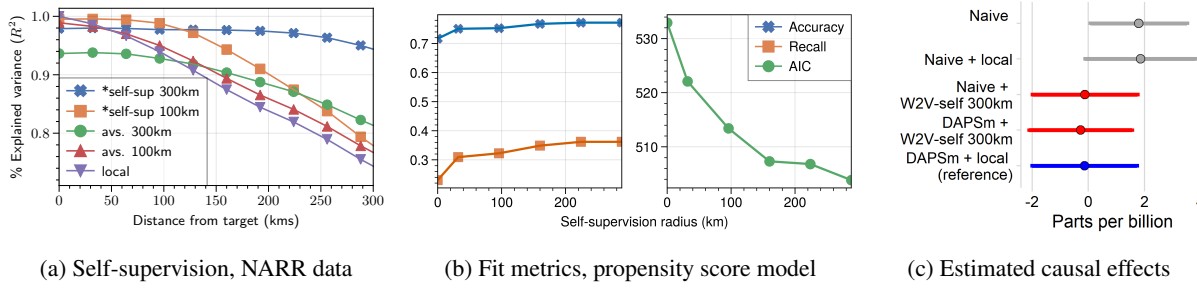

(a) Self-supervision, NARR data     (b) Fit metrics, propensity score model     (c) Estimated causal effects

Figure 3: Application 1: The effectiveness of catalytic devices to reduce power plant ozone emissions.

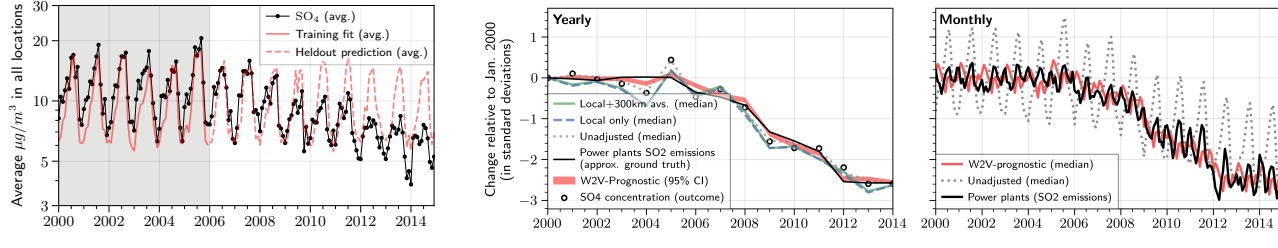

(a) Prognostic score fit averaged over the entire grid $\mathbb{G}$.

(b) Detrended series at $\mathbb{S}$ resembles power plant emissions. (*Left*) Yearly trend $\delta_{\text{year}(t)}$. (*Right*) Monthly trend $\delta_{\text{year}(t)} + \gamma_{\text{month}(t)}$

Figure 4: Application 2: Meteorological detrending of $SO_4$.

tices and merely aim to evaluate whether augmenting this approach with the *weather2vec* representation of the prognostic score offers improvement. The model is as follows:

$$\log(Y_{s,t}) \sim N(\alpha + \delta_{\text{year}(t)} + \gamma_{\text{month}(t)} + \textstyle\sum_{j=1}^{p} \beta_p X_{st}^p, \sigma^2) \tag{3}$$

for all $s \in \mathbb{S}^*, t = 1, \ldots, T$, where $\{\delta_\ell : \ell = 2000, \ldots, 2014\}$ is the year effect, $\{\gamma_\xi : \ell = 1, \ldots, 12\}$ is the seasonal (monthly) effect, $\mathbb{S}^* \subset \mathbb{S}$ are the locations of the power plants, $X_{st}^p$ are the controls with linear coefficients $\beta_p$. These controls are obtained from a B-spline basis of degree 3 using: 1) local weather only, and 2) local weather plus the *weather2vec* prognostic score. The model is fitted using Bayesian inference with MCMC. Figure 4b shows the fitted (posterior median) yearly and monthly trends. The adjusted trends resemble the power plant emissions trends much more closely than the predicted trends from models that include local or neighborhood average weather. Note in particular the "double peak" per year in the monthly power plant emissions (owing to seasonal power demand), which is captured by the detrended *weather2vec* series but not by the unadjusted one.

# 6    DISCUSSION AND FUTURE WORK

While notions NLC have been acknowledged in causal inference, potential-outcomes formalization of NLC and flexible tools to address it are lacking. We offer such a formalization,

along with a flexible representation learning approach to account for NLC with gridded covariates and treatments and outcomes measured (possibly sparsely) on the same grid. Our proposal is most closely tailored to problems in air pollution and climate science, where key relationships may be confounded by meteorological features, and promising results from two case studies evidence the potential of *weather2vec* to improve causal analyses over those with more typical accounts of local weather. A limitation of the approach is that the learned *weather2vec* representations are not as interpretable as direct weather covariates and using them could impede transparency when incorporated in policy decisions. Future work could explore new methods for interpretability. Other extensions could include additional data domains, such as graphs and longitudinal data with high temporal resolution. The links to causal interference explored in Section 4 also offer clear directions for future work to formally account for NLC in the context of estimating causal effects with interference and spillover.

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

# APPENDIX

## A   SIMULATION STUDY

The simulated data mimics the meteorological data in our applications and the matches setup of Section 2 with SUTVA and NLC. We briefly describe the different aspects of the simulations, including additional visualizations and details in Appendix G.

*Data simulation and basic linear task.* The covariates $\boldsymbol{X}_s$ are the gradient field (the first differences along rows and columns) of an unobserved Gaussian Process [Rasmussen, 2003] defined over a $128 \times 256$ grid. To fix ideas, the simulation is carried out to roughly mimic a study of pollution sources where pollution is dispersed in accordance with non-local weather covariates, so, $\boldsymbol{X}_s = (\boldsymbol{X}_s^1, \boldsymbol{X}_s^2)$ can be roughly interpreted as "wind vectors". The treatment assignment probability (the propensity score) and the outcome are computed as a "non-local" function of $\boldsymbol{X}_s$, simulated to correspond to higher probability of treatment in areas that tend to disperse more pollution. Such an assignment can be performed using a convolution operation. More precisely, let $\mu = \sum_{j \in \{1,2\}} K_j \star \mathbf{X}^j$ be the result of convolving $\mathbf{X}$ with a specially designed convolution kernel $K$. Then, the treatment assignment probability is $A_s \sim \text{Bernoulli}(\mu_s)$ and the outcome is $Y_s = -\mu_s + \epsilon_s + \tau A_s$, where $\tau$ is the treatment effect and $\epsilon_s$ is a mixture of spatial and random noise of unit variance. $\tau = 0.1$ in all experiments. $K$ has dimensions $13 \times 13$ (its size determines the radius of NLC). $K_1$ contains -1's in the upper half, +1's in the lower half, and 0's in the middle row. $K_2 = K_1^\top$. Convolving a a gradient field with $K$ is an approximate form of identifying valleys and hills in the potential of the gradient field. In this basic formulation, $\mathbb{S} = \mathbb{G}$, meaning that $A_s$ and $Y_s$ are densely available over the grid $\mathbb{G}$.

*Causal estimation procedure.* We first estimate a propensity score model $\hat{\mu}_s$ using the learned $\boldsymbol{Z}_{\theta,s}$. For the supervised variant, $\hat{\mu}_s = \text{sigmoid}(\boldsymbol{Z}_{\theta,s})$. But for the self-supervised, it is constructed from an feed-forward network with one or two hidden layers (see the Appendix for details). In all cases, the final estimate of $\tau$ is produced using the inverse probability weighting (IPW) estimator [Cole and Hernán, 2008]

$$\hat{\tau}_{\text{IPW}} = |\mathbb{S}|^{-1} \sum_{s \in \mathbb{S}} \{ (Y_s / \hat{\mu}_s) \mathbb{I}(A_s = 1) \\ - (Y_s / (1 - \hat{\mu}_s)) \mathbb{I}(A_s = 0) \}.$$

Although more sophisticated causal estimators could be used, the exercise only intends to measure the degree to which a propensity score anchored to $\boldsymbol{Z}_{\theta,s}$ encodes the necessary NLC information.

*Additional task variants.* In one variant, we consider a sparse configuration in which the outcome and treatment are only sparsely available in a subset $\mathbb{S}$ of 500 randomly selected points in $\mathbb{G}$. In another variant, we evaluate the results on a *non-linear* version of the treatment assignment logits, computed as $\mu = \sum_{j \in \{1,2\}} K_j \star \text{sign}(\mathbf{X}^j)$. This small amount of non-linearity strongly increases the complexity of the problem. Both tasks variants are also combined, resulting in 4 total tasks.

*Baselines.* Four baselines are considered for estimating the propensity score: no controls (Unadjusted); controlling for local covariates (Local only); controlling for local and averages of neighboring covariates (Local + Averages) – assuming the neighborhood size 13 of NLC is known; and a purely spatial random effects model for the treatment (Spatial RE only), specified as a conditional auto-regression (CAR) model with Bernoulli likelihood [Besag, 1974] (more details in the Appendix). We compare against the self-supervised and supervised versions of *weather2vec*, including a supervised version combined with the spatial RE.

*Results.* A total of 10 experiments are performed for each configuration and task. The results are shown in Figure 5. For the dense case when $\mathbb{S} = \mathbb{G}$, the supervised *weather2vec* outperforms all other methods (panels a and b), exhibiting near 0 bias in the linear case and a small amount of finite-sample bias for the nonlinear task owing to the relatively small number of observations. We see analogous performance for the unsupervised *weather2vec* for the case when $\mathbb{S}$ has only 500 locations (panels c and d), noting further the poor performance of the supervised *weather2vec* in this case owing to overfitting the very small number of observations. Note that the spatial RE only model outperforms the use of local and average neighbors, but its addition to *weather2vec* deteriorates performance.

## B   SPATIAL STATIONARITY AND THE U-NET FOR SUMMARIZING NON-LOCAL DATA

With an input grid consisting of $\boldsymbol{X}_s \in \mathbb{R}^d$, the U-net transformation involves two parts: a contractive stage and an almost symmetric expansive stage. Both of these steps use convolutions with learnable parameters and non-linear functions to aggregate information from the input grid spatially and create rich high-level features. The convolutions in the contractive path duplicate the number of latent features at each layer. Then, these intermediate outputs go through *pooling* layers which halve the spatial dimensions. Together, these operations augment the dimensionality of each point of the grid, combining information at many spatial points to richer information contained at fewer points. Convolutions propagate information spatially, and the deeper they are in the contractive path, the larger their propagation reach (in the original scale of the input grid). So, as the spatial dimension gets reduced, each element of the reduced grid contains more information from the original spatial scale.

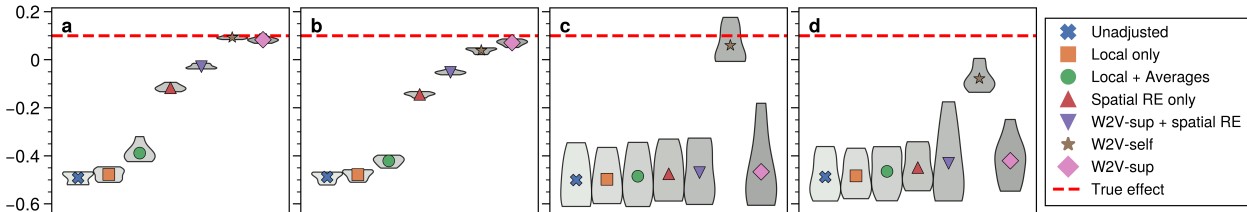

Figure 5: Results from the simulation study. (a) the basic linear task. (b) the non-linear task. Both of these tasks use outcomes and treatments densely available on the grid ($\mathbb{S} = \mathbb{G}$). (c) and (d) are the same tasks respectively, but only 500 random locations are in $\mathbb{S}$.

The expansive path, on the other hand, uses *up-sampling* to progressively interpolate the deep higher-level features back to a finer spatial lattice, and then uses convolutions to reduce back the latent dimensionality at each grid point; with the characteristic that, in contrast to the input grid, every point now localizes spatial information. The output vector can have any arbitrary dimension after possibly applying an additional linear or convolutional layer followings the expansive path (or before the contractive path, or both). The result is an output grid of the same dimension of the input grid, where each point contains a vector $Z_{\theta,s} \in \mathbb{R}^k$ that encodes non-local information from features observed across many points in the original input grid. The unknown weights $\theta$ dictate the size – the "radius of influence" – and what non-local information is summarized by $Z_{\theta,s}$. The weights will include in its computations values of $X_s$ that are highly predictive in the specified learning task. We have just provided an intuitive explanation of the U-net's computational model; refer directly to Ronneberger et al. for details [Ronneberger et al., 2015].

In summary, the essential properties of U-nets to parameterize representation maps for NLC are: spatial stationarity, automatic determination of the effective radius of influence, and efficient detection of spatial patterns. Notice that U-nets are not the only neural network architecture with this property. Moreover, many variants of the basic U-net have appeared in recent years in the literature [Siddique et al., 2021]. Most of those improvements are compatible with the ideas presented in the rest of this paper.

## C  PROOFS

**Proof of Proposition 1.** For convenience, drop the subscript $s$ and boldface notations, and denote $L^c = L' \setminus L$. We will use a graphical argument based on the backdoor criterion [Pearl, 1988, ch. 4.3]. Suppose that $L^c \to A$ (here $\to$ means causation) and observe the two following facts: first, a path $Y \to L^c \to A$ would violate the assumption of pre-treatment covariates; second, a path $Y \leftarrow L^c \to A$ would need to be absent or be blocked by $L$ due to sufficiency. If blocked, it must be of the form $Y \leftarrow L \leftarrow L^c \leftarrow A$

since a reversed first arrow would violate the pre-treatment assumption, implying $L^c \perp\!\!\!\perp Y \mid L$ (and as a consequence, conditionally independent of $Y(0), Y(1)$). An analogous argument shows that assuming $L^c \to Y$ would imply $L^c$ is conditionally independent from $A$ given $L$. In summary, conditioning on $L^c$ does not open any new (backdoor) paths from $A$ to $Y$. And the result follows from the backdoor criterion. $\qquad \square$

**Proof of Proposition 2.** This is a standard result in introductory expositions of potential outcomes. For each $a \in \{0, 1\}$ we have that

$$\mathbb{E}[\mathbb{E}[Y_s \mid L_s, A_s = a]] = \mathbb{E}[\mathbb{E}[Y_s(a) \mid L_s, A_s = a]]$$
$$= \mathbb{E}[\mathbb{E}[Y_s(a) \mid L_s]]$$
$$= \mathbb{E}[Y_s(a)].$$

The first equality follows from SUTVA; the second from sufficiency; the third from the law of iterated expectation. Finally, $\mathbb{E}(Y_s(a)) = |\mathbb{S}|^{-1} \sum_s Y_s(a)$ by definition, implying the proposition's statement. $\qquad \square$

**Proof of Proposition 3.** We follow Hansen [2008]'s formulation of the prognostic score, and prove the results along the lines of [Rosenbaum and Rubin, 1983, theorems 1-3]. We'll proceed in three steps. All which are somewhat informative of the role of the prognostic score. Again, we drop the subscript $s$ and boldface from the notation for clarity.

*Step 1. Conditional expectation of the outcome is a prognostic score.* Denote $\psi(L) = \mathbb{E}[Y(0) \mid L]$. We want to show the balancing property: $Y(0) \perp\!\!\!\perp L \mid \psi(L)$.

Recall the definition of conditional expectation (see [Williams, 1991, ch. 9.2]): $Z = \mathbb{E}[Y \mid L]$ iff $\mathbb{E}[Y\mathbb{I}(L \in A)] = \mathbb{E}[Z\mathbb{I}(L \in D)]$ for any $L$-measurable set $D$. We will use this definition and show that $\Pr(Y(0) \in C \mid L) = \Pr(Y(0) \in C \mid \psi(L))$, implying the required independence. (Conditioning on $(L, \psi(L))$ is equivalent to only condition on $L$.)

Now, since $\psi(L)$ is a function of $L$, the event $\psi(L) \in D$ can be re-written as $L \in \psi^{-1}(D)$ using the pre-image notation.

Then,

$$
\begin{aligned}
\mathbb{E}[\mathbb{I}(Y(0) \in C)&\mathbb{I}(\psi(L) \in D)] \\
&= \mathbb{E}[\mathbb{I}(Y(0) \in C)\mathbb{I}(L \in \psi^{-1}(D))] \\
&= \mathbb{E}[\mathbb{E}[(Y(0) \in C) \mid L]\mathbb{I}(L \in \psi^{-1}(D))],
\end{aligned}
$$

implying that $\mathbb{E}[\mathbb{I}(Y(0) \in C) \mid L] = \mathbb{E}[\mathbb{I}(Y(0) \in C) \mid \psi(L)]$. The result then follows from noting that probabilities are expectations of indicator functions.

*Step 2. Any other prognostic score $b(L)$ is finer than $\psi(L)$.* Suppose it is not the case, then there are $\ell_1, \ell_2$ such that $\psi(\ell_1) \neq \psi(\ell_2)$ but $b(\ell_1) = b(\ell_2)$. But by the balancing property we have that

$$
\begin{aligned}
\mathbb{E}[Y(0) \mid b(L) = b(\ell_1)] \\
= \mathbb{E}[Y(0) \mid b(L) = b(\ell_1), L = \ell_1] \\
= \psi(\ell_1),
\end{aligned}
$$

which would imply that $\mathbb{E}[Y(0) \mid b(L) = b(\ell_1)] \neq \mathbb{E}[Y(0) \mid b(L) = b(\ell_2)]$, violating the assumption that $b(\ell_1) = b(\ell_2)$ and leading to a contradiction. Thus $\psi(\ell_1) = \psi(\ell_2)$ implies that $b(\ell_1) = b(\ell_2)$, which in turn implies the existence of some function $\psi(L) = f(b(L))$ and thus $\psi(L)$ is coarser.

*Step 3. If $b$ is a prognostic score, then $L$ is sufficient iff $b(L)$ is also sufficient.* First, if $b(L)$ is sufficient, then the proof is trivial. So let's consider the opposite case. First we show that $\Pr(Y(0) \in C \mid A, b(L)) = \Pr(Y(0) \in C \mid b(L))$.

The proof follows from the following identities

$$
\begin{aligned}
\Pr(Y(0) &\in C \mid A, b(L)) \\
&= \mathbb{E}[\mathbb{I}(Y(0) \in C) \mid b(L)] \\
&= \mathbb{E}[\mathbb{E}[\mathbb{I}(Y(0) \in C) \mid L] \mid A, b(L)] \\
&= \mathbb{E}[\mathbb{E}[\mathbb{I}(Y(0) \in C) \mid \psi(L)] \mid A, b(L)] \\
&= \mathbb{E}[\mathbb{I}(Y(0) \in C) \mid \psi(L)] \\
&= \Pr(Y(0) \in C \mid b(L)).
\end{aligned}
$$

The first equality is by definition, the second by iterated expectation, the third one by the sufficiency of $L$; the fourth one is because $\psi(L)$ is balancing (Step 1); the fifth one is because $\psi(L)$ is a function of $b(L)$ by Step 2; the last one is by definition.

Finally, for the treated outcome $Y(1)$, the assumption of no effect modification means that the same argument carries on for $Y(1)$ (since $Y(1) - Y(0)$ is independent of $A$). $\quad\square$

**Proof of Proposition 4.** Let $a \in \{0, 1\}$. By the assumption of conditional independence of the treatments given $\boldsymbol{L}_s$ (assumption (2) in the proposition), we have that

$$
\mathbb{E}[Y_s \mid \boldsymbol{L}_s, A_s = a]] = \mathbb{E}[Y_s \mid \boldsymbol{L}_s, A_s = a, \mathbf{A}_{\mathcal{N}_s \setminus \{s\}}]
$$

Having noted this, the proof is identical to that of Proposition 2

$$
\begin{aligned}
\mathbb{E}[\mathbb{E}[Y_s &\mid \boldsymbol{L}_s, A_s = a, \mathbf{A}_{\mathcal{N}_s \setminus \{s\}}]] \\
&= \mathbb{E}[\mathbb{E}[Y_s(a_s = a, \mathbf{A}_{\mathcal{N}_s \setminus \{s\}}) \mid \boldsymbol{L}_s, A_s = a, \mathbf{A}_{\mathcal{N}_s \setminus \{s\}}]] \\
&= \mathbb{E}[\mathbb{E}[Y_s(a_s = a, \mathbf{A}_{\mathcal{N}_s \setminus \{s\}}) \mid \boldsymbol{L}_s]] \\
&= \mathbb{E}[Y_s(a_s = a, \mathbf{A}_{\mathcal{N}_s \setminus \{s\}})]
\end{aligned}
$$

where the first identity is due to SUTNVA; the second one is by neighborhood-level sufficiency (assumption (1) in the proposition); and the third one is by the law of iterated expectation. Finally, $E[Y_s(a_s = a, \mathbf{A}_{\mathcal{N}_s \setminus \{s\}})] = (1/|\mathbb{S}|) \sum_s Y_s(a_s = a, \mathbf{A}_{\mathcal{N}_s \setminus \{s\}})$ since the randomness in the expectation is due to $s$ uniformly from $\mathbb{S}$. $\quad\square$

# D   MOTIVATING EXAMPLE FOR THE SELF-SUPERVISED MODEL: PERFECT ENCODING

Assume that the covariates $\boldsymbol{X}_s$ have dimension $d = 1$ and that the self-supervision task is to learn the adjacent values in the grid (north, west, south, east) and the central point $s = (i, j)$ using the representation $\boldsymbol{Z}_{\theta,s}$. If we set the representation dimension to $k = 5$, then the obvious candidate for the representation is

$$
\boldsymbol{Z}_{(i,j)} = (\boldsymbol{X}_{(i-1,j)}, \boldsymbol{X}_{(i,j-1)}, \boldsymbol{X}_{(i+1,j)}, \boldsymbol{X}_{(i,j+1)}, \boldsymbol{X}_{(i,j)})^\top
$$

Now let $\boldsymbol{\gamma}(\ell) = (\boldsymbol{\gamma}(\ell)_1, \ldots, \boldsymbol{\gamma}(\ell)_5)$ be the $\ell$-th indicator vector with $\boldsymbol{\gamma}(\ell)_j = \mathbb{I}(j = \ell)$. Then

$$
\boldsymbol{\gamma}(1)^\top \boldsymbol{Z}_{(i,j)} = \boldsymbol{X}_{(i-1,j)}, \cdots \qquad \boldsymbol{\gamma}(5)^\top \boldsymbol{Z}_{(i,j)} = \boldsymbol{X}_{(i,j)}
$$

Hence $\boldsymbol{Z}$ is a perfect encoding and the $\boldsymbol{\gamma}(\ell)$'s are perfect classifiers for each offset $\ell$. To generalize this idea to higher dimensions $d > 1$, we can take $\boldsymbol{\gamma}(\ell)$ to be a $d \times k$ matrix for each $\ell$. Then $\boldsymbol{\gamma}(\ell)^\top \boldsymbol{Z}_{\theta,s}$ is a $d$-dimensional vector for each offset $\ell$. The same idea is behind the self-supervised model, which takes $\boldsymbol{\Gamma} = \gamma^\top$ as a $k \times k$ matrix and adds a decoder neural network. Rather than using indicator functions for $\Gamma$, the method formulates it as a neural network that is a function of the offset.

# E   A CONNECTION BETWEEN THE SELF-SUPERVISED MODEL AND PCA

Principal components analysis (PCA) is closely related to a special case of the self-supervised *weather2vec* when using a single $(2R+1) \times (2R+1)$-convolution instead of the U-net, leaving $h_\psi$ as the identity function, and defining the offset embedding $\boldsymbol{\Gamma}(\delta)$ as independent $d \times k$ vectors for each offset $\delta$ (rather than a neural network with $\delta$ as a continuous input). The equivalence is in the sense of reconstruction since both methods can be seen as minimizing the reconstruction error. However, in the self-supervised case there is no guarantee that the latent dimensions of $\boldsymbol{Z}_{\theta,s}$ will be orthogonal as in PCA applied to each patch of size $(2R + 1) \times (2R + 1)$.

## F   SOFTWARE AND HARDWARE

We use open-source software PyTorch 1.10 Paszke et al. [2019] on Python 3.9 Van Rossum and Drake Jr [1995] for training all the models on a single laptop with an Nvidia GPU 980M (8GB) and a CPU Intel i7-4720HQ at 2.60GHz. The code uses fairly standard functions for NN training and we did not attempt to optimize it for speed. We also use R 3.6 R Core Team [2021] for downloading and pre-processing atmospheric data from NARR, as well as for comparison with DAPSm in application 1 (see Appendix H).

The code for Bayesian inference in Application 2 is implemented in pure Python as a straightforward Gibbs sampler since the model is Gaussian.

## G   ADDITIONAL DETAILS OF THE SIMULATION STUDY

*Details on the simulation data.* Figure 6 illustrates the data used in the simulation study. (a) shows the simulated co-variates $\mathbf{X} = (\mathbf{X}^1, \mathbf{X}^2)$ as the gradient vector field of an unobserved potential function (sampled from a Gaussian Process) $\mathbf{F}$, whose level curves are overlay the covariate (vector field) represented by arrows. (b) and (c) jointly compose the kernel used to generate the confounding factor. (d) is the resulting treatment assignment probability for the linear task, and (e) is the corresponding non-linear variant. In both, convolutions approximately correspond to valleys and hills of the unobserved potential.

*Details on the neural network architectures.* The NNs used in the study are very lightweight, since the data consists of only one image of $128x256$. Typical NN sizes with millions of parameters would easily over-fit to this task. The basic U-net architecture used for *weather2vec* is in Figure 7, but using the simulated gradient fields instead of atmospheric covariates. All convolutions and linear layers are followed by batch normalization and SiLU activations Elfwing et al. [2018], except in the last layer.

- **Supervised *weather2vec*.** The propensity score model uses two hidden units and depth 2. The model has 1.2k (trainable) parameters.

- **Self-supervised *weather2vec*.** The auto-encoder uses 16 hidden units and depth two. The offset model $\mathbf{\Gamma}_\phi$ is a two-layer feed-forward network with 16 hidden units. The decoder $h_\psi$ is feed-forward network with one hidden layer of also 16 units. In total, the auto-encoder has 77k parameters. In addition, the propensity score model uses a feed-forward network with two hidden layers of 16 units, resulting in 600 parameters.

- **Local and local+avgs.** These baselines use the same propensity score model as the self-supervised one. Due to their smaller input size, they have around 400 pa-

rameters.

- **Spatial RE**. Rather than a neural network, we used a conditionally auto-regressive (CAR) Besag [1974] model such that $A_s \sim \text{Bernoulli}(\text{sigmoid}(\mathbf{Z}_{\theta,s}))$, $\mathbf{Z}_{\theta,s} \sim \text{CAR}(\lambda)$ and $\lambda \sim \text{Gamma}(1,1)$. The CAR portion of the negative loglikelihood penalizes the (squared) differences of adjacent values of $\mathbf{Z}_{\theta,s}$ in the grid by a factor of $\lambda$. Notice that $\lambda$ here is learned along with the model. We remark that CAR models are more scalable alternatives to Gaussian process for applications requiring only smoothing and interpolation.

- **Supervised *weather2vec* + spatial RE**. This variant formulates the representation as $\mathbf{Z}_{\theta,s} = \tilde{\mathbf{Z}}_{\theta,s} + \xi_s$, where $\tilde{\mathbf{Z}}_{\theta,s}$ is the output of the U-net and $\xi_s$ has a CAR prior and is restricted to $\sum_s \xi_s = 0$ for identifiability. Intuitively, the term $\xi_s$ captures the errors in the propensity score model that have a strong spatial distribution.

Finally, the unadjusted baseline is simply the difference of the averages of the observed treated and untreated units.

*Details on the training procedures and hyper-parameters.* In all cases, we use a fixed learning rate of $10^{-4}$, a weight decay of $10^{-4}$, and 20,000 gradient steps with the ADAM optimizer [Kingma and Ba, 2014]. The full simulation study takes about 8 hours to finish running two baselines in parallel. The values of weight decay, training epochs and learning rate were chosen as reasonable values without much additional optimization. The number of layers and architectures were chosen by inspection after a few runs, aiming to find a model small enough as to avoid over-fitting without requiring tuning the regularization hyper-parameters or early stopping.

## H   ADDITIONAL DETAILS OF APPLICATION 1

*Atmospheric data download*. The NARR [Mesinger et al., 2006] dataset associated with the application can be downloaded via FTP with the R script provided in the code accompanying the paper. The data is also publicly available for download from the website of the National Oceanic and Atmospheric Administration (NOOA) Oceanic and Administration. We could not find any license information for the dataset. We could not find any license attached to the dataset.

*Power plants data*. Information for the largest 473 coal-fired power plants emitting $SO_2$ during 2000–2014 was obtained from Papadogeorgou [2016] (publicly available under creative commons license CC0 1.0).

*Neural network architecture for self-supervised features*. The auto-encoder uses 32 hidden units and depth 3. The offset model $\mathbf{\Gamma}_\phi$ is a two-layer feed-forward network with

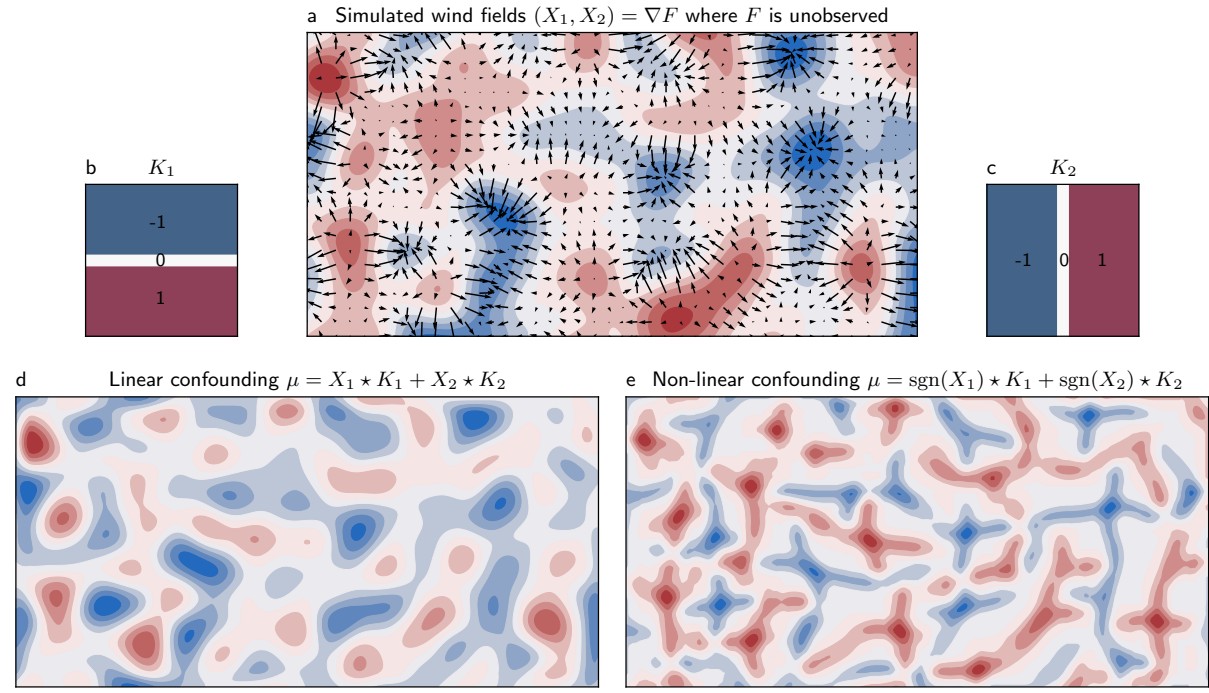

Figure 6: Simulations, components and variantes in the simulation study

32 hidden units. The decoder $h_\psi$ is feed-forward network with one hidden layer of also 32 units. In total, the auto-encoder has 1.2M parameters. The architecture is shown in Figure 7. Convolutions are followed by FRN normalization layers Singh and Krishnan [2020] and SiLU activations. Pooling uses *MaxPool2d* and upsampling use *Bilinear Up-sampling2d* as implemented in PyTorch. The model architecture was not tuned since the model with 32 hidden units seemed to work well.

*Details on the training procedures and hyper-parameters.* The model is trained for 300 epochs using batch size 4, a linear decay learning rate from $10^{-2}$ to $10^{-4}$ using the ADAM Kingma and Ba [2014] optimizer (no weight decay). The number of epochs and learning rate were tuned by inspection after a few runs simply to ensure the model was learning at a reasonable speed, but not tuned otherwise. The atmospheric covariates were standardized before training and we use a standard quadratic loss. (Using this loss is equivalent to fixing $\Sigma$ as the identity function in equation 2.)

We do not split in training and validating datasets since the model is a compression/dimensionality reduction technique, and thus it cannot over-fit. (In fact, an "over-fitting" here would be a desirable property, since it would mean a perfect dimensionality reduction.)

*Computation of the explained variance ($R^2$).* The traditional $R^2$ is defined as one minus the ratio of sum-of-squares between the prediction errors and the centered targets. Since the covariates are standardized, the latter quantity is simply

$N$. In each training epoch we collect the sum of squared prediction errors for all time periods. Denote this quantity as $SSE_j$ where $j$ indicates the covariate dimension for $j = 1, ..., d$. Then $R_2 = 1 - (Nd)^{-1} \sum_j SSE_j$ is the proposed estimator of the fraction of the variance explained.

*Comparison with DAPSm.* We modified the DAPSMm authors implementation from Github [Papadogeorgou, 2001] (no license provided) to include the *weather2vec* self-supervised features as another predictor in their otherwise unchanged propensity score model. The modified R script is in the code accompanying this paper.

# I ADDITIONAL DETAILS OF APPLICATION 2

*Neural network architecture for supervised features.* The U-net architecture for the prognostic score model is almost identical to the auto-encoder of Application 1, but with having dimension one in the last layer.

*$SO_4$ data download.* We downloaded the dataset the $SO_4$ grid for inland US from the website of the Atmospheric Composition Analysis Group's van Donkelaar et al. [2021] website Group [2001]. We could not find any license information for the dataset. Instructions for replications are provided in the code.

*Missing data.* Data for some observations in the $SO_4$ grid are missing and a few have clearly erroneous (near infinite)

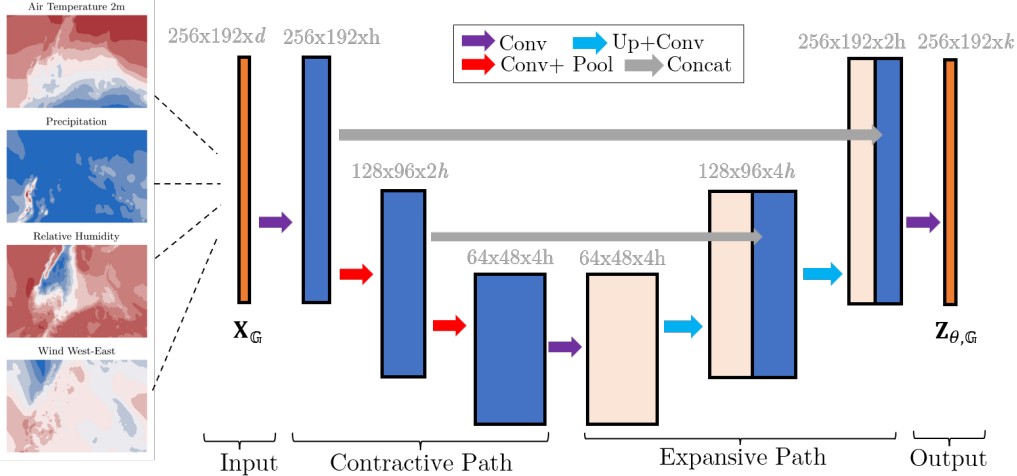

Figure 7: Basic U-net architecture used in the two applications and the simulation study.

values. In addition, some locations have data but present zeros through the entire period. We excluded these values using a binary mask in the likelihood by removing non-finite values and keeping only locations with positive observations throughout. Doing so greatly improved the quality of the fitted model. The final locations cover most of the inland U.S., with missing areas mostly outside the U.S, oceans, or the Rocky West and Great Basin.

*Details on the training procedures and hyper-parameters.* The hyper-parameters are also the same as in the self-supervised model except that we use a weight decay of $10^{-4}$ to reduce over-fitting. We did not tune this parameter, however, we did not notice a significant difference by increasing or decreasing its value by a factor of 10.

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
