# OpenReview forum: "Weather2vec: Representation Learning for Causal Inference with Non-Local Confounding in Air Pollution and Climate Studies"
_auai.org/UAI/2022/Workshop/CRL — CRL@UAI 2022 Poster_

### Official Review · Reviewer_691x · 2022-06-25
**A very solid use of neural networks and representation learning to solve a causal estimation problem**

**Rating:** 8
**Confidence:** 3

**Review:**

Reviewer summary:
This paper introduces the Non-Local Confounding (NLC) problem and contrasts it with the similar problem of interference in the potential outcomes framework. In short, the problem arises when a unit's outcome and treatment depend on the --covariates-- of close units, as opposed to interference where only the unit's outcome depends on the --treatment-- of close units. Based on this theory, the authors propose using the U-net architecture to find the adjusted effects. Their framework allows estimating the effects both in a supervised (if enough data is available) or in an unsupervised way. The authors test their method empirically, showing the effectiveness of the proposed approach.

Comments:
- The paper is well written and theoretically sound.
- The paper solves a problem that might arise in real-world scenarios. The proposed approach is interesting and relevant for the workshop.
- Is the choice of a Gaussian distribution in (2) ad-hoc? This is unclear from the text, and I suppose one needs to model the covariates well by choosing an appropriate likelihood function. If this is the case, though, I can see how the proposed approach is a bit more complicated to use because defining the likelihood for a high-dimensional variable is not trivial. Can you please comment on that?
- In addition to the problems of interpretation, what are other limitations or modes of failure of the proposed approach? Did you somehow stress-test the models?
- I am interested in the power of the representations learned by the model. I conjecture these representations are appropriate only for a subset of the possible tasks on the type of data. Otherwise, the representation would have to be the data itself, which would beat the purpose of representation learning. Do you know which type of tasks the learned representations would --not-- be appropriate for (maybe in the context of weather data, your area of expertise)? Most importantly, is this even an interesting question?

---

### Official Review · Reviewer_JLWB · 2022-06-28
**Balancing scores for non-local confounding**

**Rating:** 5
**Confidence:** 4

**Review:**

The paper tackles an interesting issue that arises in causal inference with spatial data or certain kinds of network dependencies.

Some parts of the paper are not really clear or fully developed. In particular, the solution proposed using U-nets is quite unclear to me. Section 3.2 seems to be where the main substantive proposal is, but this section is far too brief and not clearly written. I don't really understand what role the U-net architecture plays or what assumptions underlie the approach (i.e., when it would break down, what formal guarantees or properties it has, etc.). I'm not really clear on how NLC is being accounted for in the balancing score or combined with DAPSm.

I also think the description of interference (and thus the supposed contrast with NLC) is not quite fair or fully representative of that literature. It is true that one kind of interference phenomenon that people study is the effect of one unit's exposure on another unit's outcome. But also interference includes the phenomenon where outcomes are dependent across units, and even covariates. There are many different mechanisms that go under the heading of interference, including contagion and homophily. (See, e.g., Shalizi and Thomas, "Homophily and contagion are generically confounded..." in Soc Methods & Research 2011 and Ogburn, Shpitser, and Lee "Causal inference, social networks, and chain graphs" in JRSSA 2020. In Bhattacharya et al "Causal inference under interference and network uncertainty" UAI 2019, the authors use chain graph models to represent different interference hypotheses.) Though I'm not sure I've seen much work specifically about on unit's covariates affecting other units (at least not under the name "NLC"), this does appear in the interference literature, for example in twin studies where pairs have shared covariates (same household-level features). This is not to say that there is no new contribution here, but just that the relationship to existing interference literature could be clarified and improved.

---

### Meta-Review · Program_Chairs · 2022-07-06

**Recommendation:** Accept (Poster)
**Confidence:** 4

**Metareview:**

This is an interesting approach using crl for real world applications, should be discussed at the workshop.

---

### Decision · Program_Chairs · 2022-07-06

Accept (Poster)